# Topology of vibrational modes predicts plastic events in glasses

Zhen Wei Wu ®[1,2] ✉, Yixiao Chen[3], Wei-Hua Wang ®[4], Walter Kob ®[5] ✉ & Limei Xu[2,6,7] ✉

The plastic deformation of crystalline materials can be understood by considering their structural defects such as disclinations and dislocations. Although also glasses are solids, their structure resembles closely the one of a liquid and hence the concept of structural defects becomes ill-defined. As a consequence it is very challenging to rationalize on a microscopic level the mechanical properties of glasses close to the yielding point and to relate plastic events to structural properties. Here we investigate the topological characteristics of the eigenvector field of the vibrational excitations of a two-dimensional glass model, notably the geometric arrangement of the topological defects as a function of vibrational frequency. We find that if the system is subjected to a quasistatic shear, the location of the resulting plastic events correlate strongly with the topological defects that have a negative charge. Our results provide thus a direct link between the structure of glasses prior their deformation and the plastic events during deformation.

The structural disorder of glasses allows them to have a multitude of properties that are absent in crystals but which can be exploited in many technical applications[1–4]. The downside of this disorder is that it has hindered us to come up with a reliable microscopic description of many of these features since at present we lack a solid understanding how the atoms are arranged on the level of the particles[2]. Prominent examples of these properties are the specific heat and the electric conductivity of glasses at low temperatures[5,6], or the plastic deformation of the solid upon applied load before the yielding point[7]. While in crystals dislocations and disclinations allow to rationalize plastic deformation[8], the structural disorder of glasses makes it impossible to come up with a reasonable definition of a defect and as a consequence it becomes very challenging to establish a connection between the local structure and the yielding of the sample, despite a multitude of efforts[9–16]. Although the local structure of systems composed of particles that have a mesoscopic size (colloids, granular systems,...) can be determined on the level of the particles[14,17,18], the mechanical properties of such materials are dominated by particle-interactions that are

mainly of excluded volume type, i.e., very different from the ionic or covalent interactions found in atomic systems. As a consequence our understanding on how the local structure and interactions affect the mechanical properties of materials like metallic or oxide glasses is far from being satisfactory, despite the long history of research on this topic[19].

Early studies on this question showed that for simple model systems an increasing strain leads to a destabilization of the local packing and results in a strain activated, instead of thermally activated, stress relaxation[20], a picture that was elaborated by Falk and Langer who introduced the notion of shear transformation zones (STZ)[9]. Many subsequent studies have related these STZ and the plastic events (PE) to structural quantities, a high local entropy, and the local specific heat, indicating that this concept is useful[10,12,14,18,21–24].

By probing the local yield stress in the glass, Patinet et al. showed that the zones in which the local yield stress is weak correlate well with the spots at which the globally sheared sample shows a plastic event[13,25]. In particular, it was found that this purely local measure

[1]Institute of Nonequilibrium Systems, School of Systems Science, Beijing Normal University, 100875 Beijing, China. [2]International Center for Quantum Materials, School of Physics, Peking University, Beijing 100871, China. [3]Yuanpei College, Peking University, 100871 Beijing, China. [4]Institute of Physics, Chinese Academy of Sciences, Beijing 100190, China. [5]Department of Physics, University of Montpellier and CNRS, 34095 Montpellier, France. [6]Collaborative Innovation Center of Quantum Matter, Beijing 100871, China. [7]Interdisciplinary Institute of Light-Element Quantum Materials and Research Center for Light-Element Advanced Materials, Peking University, Beijing 100871, China. ✉e-mail: zwwu@bnu.edu.cn; walter.kob@umontpellier.fr; limei.xu@pku.edu.cn

shows a higher predictive power for identifying sites of plastic activity than the structural properties considered so far, thus demonstrating that PE can be predicted from local information and that therefore it is reasonable to attempt to correlate PE with local structural information[26].

At low temperatures the motion of the atoms is of vibrational nature and since PEs are related to a local mechanical instability of the particle configuration it can be expected that the latter are associated with quasi-localized soft modes[27]. Various studies support this view, showing that the location of these soft modes are often, but not always, related to the local arrangement of the particles[11,12,17,27–39].

In view of the difficulty to identify the relevant structural features that are responsible for the occurrence of PE, recent studies have used machine learning approaches to identify a connection between these two quantities[16,40–43]. Although these works have demonstrated that this technique is indeed able to predict to some extent the location of PEs, Richard and collaborators have recently shown that the quality of these predictions is not always superior to the one of simple local structural predictors and hence one concludes that at present we are still lacking insight on the relevant quantity that is able to predict PEs[15].

Although soft modes show a good correlation with plastic events, one must recall that yielding is a cooperative phenomenon. Therefore it is important to study not only the effect of individual modes, but instead to consider the total vibrational field given by the weighted sum over all the modes, since this will determine the displacement field of all the atoms. In the present work we hence focus on the eigenvector field for the different modes, and in particular study its topological properties. The idea that topological properties might be useful to understand certain thermodynamic and kinetic features of glasses has been suggested only in recent years[44,45]. Subsequently Baggioli et al. have put forward the relevance of topological properties

by showing that plasticity is mediated by topological features in the non-affine displacement field of glasses under deformation[39]. In the present work we find that the topological singularities of this field, averaged over the modes, are closely related to the plastic events, showing that the vibrational modes allow direct prediction of the location of plastic transformations if the sample is sheared.

## Results

### Structure of topological defects in the eigenvectors

The system we study is a two-dimensional binary mixture of Lennard-Jones particles, the interactions of which have been truncated at the minimum. The liquid was equilibrated at a high temperature and then cooled down, at constant volume, below the kinetic glass transition temperature. Using a conjugate gradient procedure we determined the local minimum of the potential energy and subsequently calculated the eigenvectors and vibrational frequencies $\omega$. Two samples have been produced: One with a moderate cooling rate and a second one with a slow cooling rate. More details on the potential and the simulations are given in the Methods and structural data is given in the Supplementary Figs. 1 and 2.

In order to comprehend the evolution of the morphology of the eigenvector field with $\omega$, we show in Fig. 1 the vibrational modes for different values of the frequency. (The vibrational density of states is shown in panel (f).) For low frequencies, panel (a), the mode is composed of a multitude of swirls that have roughly the same size. (In Supplementary Fig. 4 we show that at very low frequencies the eigenvector does show a regular pattern, as expected for a homogeneous elastic solid.) By calculating from the eigenvector field the topological charge, i.e., the winding number, we can identify the location of the singularities (topological defects, TD) of the field (see Methods) and these positions are included in the graph as well, with

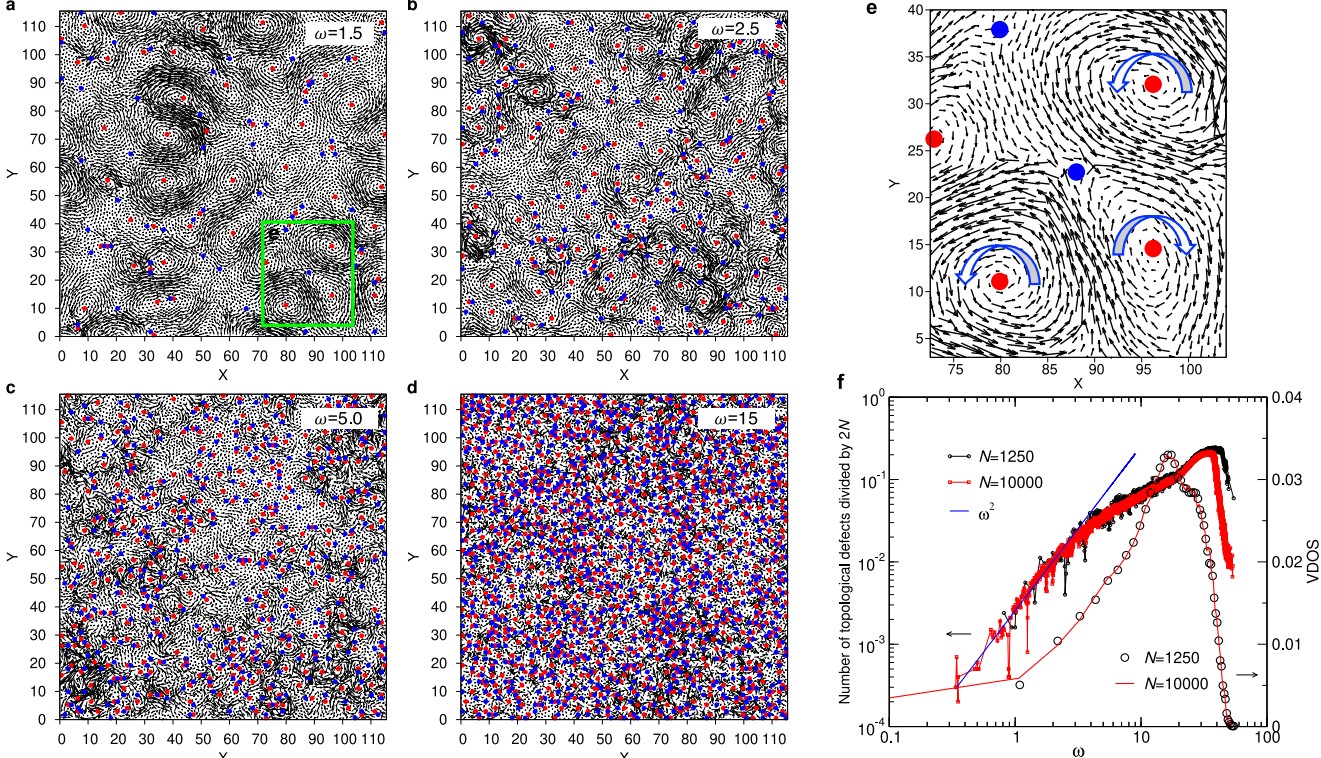

**Fig. 1 | Normal modes eigenvectors and topological defects.** Sample produced with moderate cooling rate. **a–d** Snapshots of the normalized eigenvector at $\omega = 1.5$, 2.5, 5, and 15, respectively. The green box in panel **a** shows the location of the zoom of panel **e**. Note that in **a–d** the magnitude of the eigen-vectors in different modes are shown on the same scale (amplified 150 times for visibility). **e** Four $+1$ defects (vortex, red) and two $-1$ defects (anti-vortex, blue). The big arrows mark the chirality of the defects. **f** The number of topological defects per degree of freedom, $2N$, as a function of $\omega$ (left scale) for two system sizes. The solid line is a power-law with exponent 2.0. Right scale: Vibrational density of states.

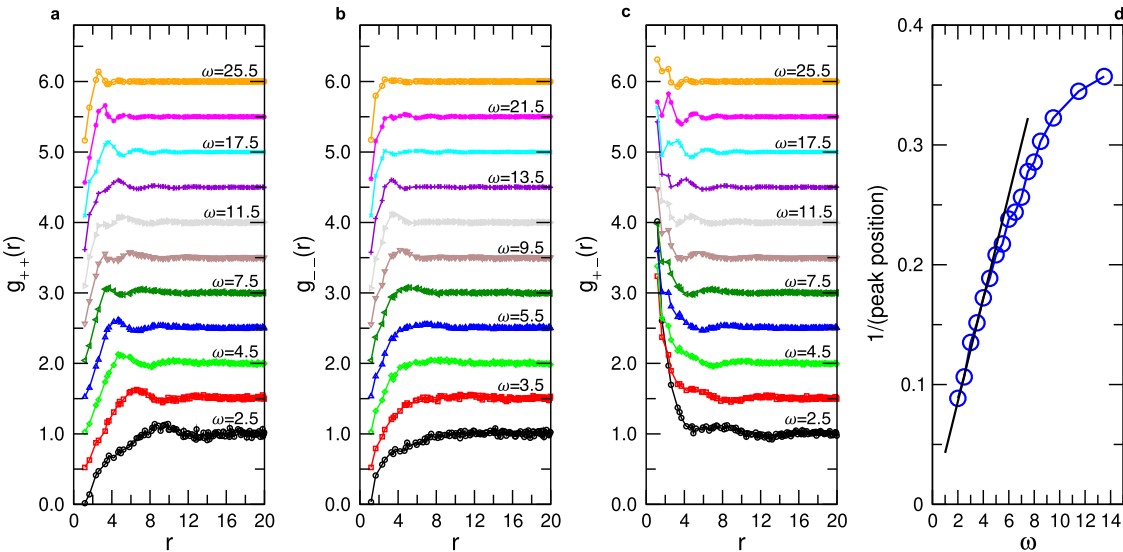

**Fig. 2 | Structure of the topological defects in the eigenmodes.** Sample produced with moderate cooling rate. **a–c**: Pair correlation functions for $+/+$, $-/-$, and $+/-$ defect pairs for different values of $\omega$. Curves for $\omega > 2.5$ are shifted upward by multiples of 0.5. **d** Inverse of the position of the first peak in $g_{++}(r)$ as a function of frequency. The straight line is a linear fit to the data at low $\omega$.

red and blue points marking charges $+1$ (vortex) and $-1$ (anti-vortex), respectively; see also panel **(e)**.

A closer inspection of Fig. 1a demonstrates that at low $\omega$ the positions of the TD are not random but correlated: While TDs with the same sign have the tendency to stay away from each other, TDs with opposite sign seem to attract each other and below we will quantify this behavior in more detail. Note that two neighboring swirls with opposite sign of phase chirality (direction of rotation of the vortex) have an interface that is not frustrated, Fig. 1e, while if they have the same sign of chirality, they are not compatible with each other and thus at their interface there will be a TD with charge $-1$, Fig. 1e.

If $\omega$ is increased, panels **b–d**, it becomes increasingly difficult to identify isolated zones in which the atoms move in an coherent manner, i.e., the swirls become ill-defined. It is, however, still possible to identify the TDs and thus to recognize that their number increases quickly with increasing frequency. Panel **f** shows the total number of TD as a function of frequency and we find that at low $\omega$ this number increases like $\omega^2$. This dependence can be understood by recalling that for a continuous homogeneous elastic medium the number of maxima/minima of a plane wave with wave-vector $\mathbf{q}$, i.e., an acoustic mode, increases like $q = |\mathbf{q}|$. If $k$ plane waves having the same $q$ but different orientations are superposed, this will give rise to $k \cdot q$ minima/maxima. Since in $D$ dimensions the number of modes increases like $k \propto q^{D-1}$, the number of minima/maxima scales like $q^D$ and because $\omega \propto q$, this rationalizes the $\omega$-dependence of the number of TD (see also Fig. 2d and Supplementary Fig. 4). At $\omega \approx 3.5$ this quadratic dependence crosses over to a weaker one, most likely because the glass can no longer be considered as an elastic medium. This observation is in harmony with the fact that at this frequency also the density of states ceases to show the linear dependence on $\omega$ expected from the Debye-law, see Supplementary Fig. 3b. This coincidence also hints a possible relation between the TD and the so-called boson peak[6], i.e., this peak might be related to the fact that on length scales smaller than the corresponding acoustic modes the notion of homogeneous elasticity is no longer applicable. The details of this possible connection should be investigated in future work.

The positional correlation observed in Fig. 1a between the defects can be quantified by means of the corresponding partial radial distribution functions $g_{++}(r)$, $g_{+-}(r)$, and $g_{--}(r)^2$, Fig. 2. For low $\omega$ one notices that $g_{++}$ has a correlation hole at small $r$, followed by a small peak at $r \approx 9$, signaling the typical distance between these TDs,

in agreement with the fields shown in Fig. 1. With increasing $\omega$ this peak shifts to smaller $r$ and one finds that at small and intermediate $\omega$, its location is proportional to $1/\omega$, see panel **d**, which is consistent with the $q$-dependence of the acoustic modes discussed above. This proportionality holds up to $\omega \approx 4.5$, at which point it crosses over to a weaker $\omega$-dependence. In this range of $\omega$, $g_{++}$ displays several wiggles, indicating that the TD with charge $+1$ form a structure that is reminiscent of the one of a hot liquid, i.e., these TDs have the tendency to repel each other. For $g_{--}(r)$, panel **b**, we find no nearest neighbor peak at low $\omega$, showing that these TD behave like an ideal gas. Only if the frequency reaches $\omega \approx 7.5$ one notices a peak at around $r = 5$, but at larger $r$ no further wiggles are observed. This indicates that negative TDs are clustered but do not form a structure that are liquid-like. If $\omega$ is increased even more, the correlation function becomes again flat (except for a correlation hole at $r = 0$), showing that at high frequencies these TDs become completely uncorrelated. These observations for $g_{--}(r)$ indicate that the relative arrangement of the $-1$ defects is significantly less structured than the one of the $+1$ TD and below we will discuss the origin of this behavior.

The correlator $g_{+-}(r)$, panel **(c)**, shows a dependence on $r$ and $\omega$ that is very different from the two other correlators. Independent of $\omega$ one finds a strong peak at small $r$'s, demonstrating that the positive and negative TD attract each other. In view of this strong correlation one can conclude that the positive TD form a liquid-like structure and that it is the attraction between the positive and negative TD which makes that the latter become correlated as well, although to a lesser extent, in agreement with the data in panel **b**.

**Correlation between topological defects and plastic events**

Since TDs are singularities in the field of the eigenvector, it can be expected that they are directly related with the heterogeneous mechanical response of the system under shear. To explore this connection we have determined the plastic instabilities of the sample if it is put under an athermal quasi-static simple shear, see Methods for details. For a sheared configuration at strain $\gamma$ we calculate $D_{min}^2$, the field of non-affine displacement of the particles between two consecutive configurations (having a difference in strain of 0.05%)[9]. After a drop in the stress, signaling that a plastic event has occurred, we identify the particles that have a high $D_{min}^2$ (top 5%, see Supplementary Fig. 6), and associate these particles to a PE. The location of these PEs can be correlated with the position of the TDs by means of a

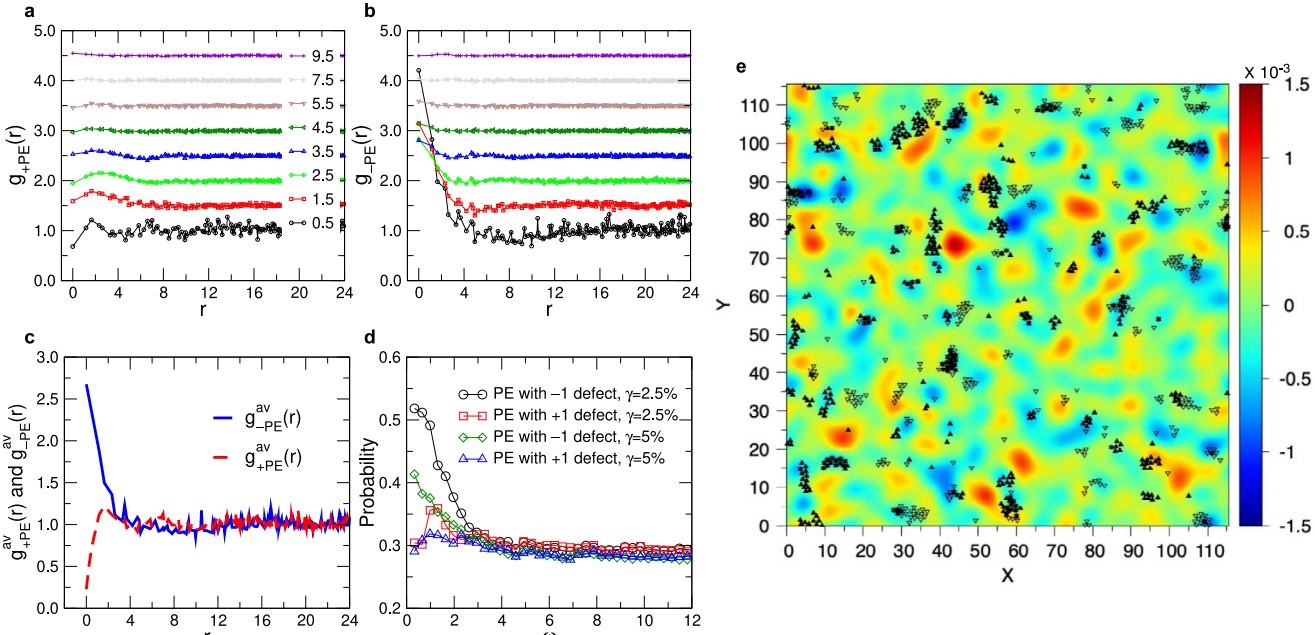

**Fig. 3 | Spatial correlation between the location of topological defects and the plastic events.** Sample produced with moderate cooling rate. **a** and **b**: Correlation functions between plastic events at strain $\gamma = 2.5\%$ and topological defects with positive charge, **a**, and negative charge, **b**, for different frequencies. **c**: Weighted sums over $\omega$ of $g_{+PE}(r)$ and $g_{-PE}(r)$. The weights are $\omega^{-2}$ and all modes up to $\omega = 3.5$ have been taken into account. **d** Probability that a PE has a distance less than 1.6 from a negative/positive PE (see text). Two values of the strain $\gamma$ are shown.

**e** Charge density field of the TDs (color map) and of the PE (symbols) at strain $\gamma = 0.025$. The density field, in units of charge per unit area, has been obtained by averaging the TDs over all frequencies up to $\omega_{max} = 3.5$, using a weight factor $\omega^{-2}$, and a subsequent Gaussian smoothing function of width 4. PE resulting from a shear in the positive and negative $x$-direction are marked by filled and open symbols, respectively.

corresponding radial distribution function $g_{\alpha PE}(r)$, where $\alpha \in \{\pm 1\}$, see Methods. In Fig. 3a, b we show the resulting correlation for different values of $\omega$ and one recognizes that $g_{+PE}(r)$ is basically flat, implying that there is only very little correlation of the PE with the positive TDs, and this holds for all frequencies. The only noticeable feature in the curve is the small peak at around $r = 2$ if $\omega$ is small, the origin of which is discussed below.

In contrast to this, $g_{-PE}(r)$ shows at small and intermediate values of $\omega$ a marked peak at small $r$, panel **b**, demonstrating that PE are closely related to low-frequency TDs with negative charge. This results rationalizes also the presence of the small peak found at $r \approx 2$ in $g_{+PE}(r)$ since from Fig. 2c we know that TDs with positive and negative charges are significantly correlated and hence the correlation between negative TDs and PEs will induce a correlation between positive TDs and PEs.

Panels **a** and **b** show that TDs and PEs are correlated if $\omega$ is less than $\omega_{max} = 3.5$, while for higher frequency there is no correlation. (Note that in panel (b) we find a non-trivial correlation even for frequencies that are not small, e.g., $\omega = 1.5$, which indicates that even under quasistatic shear conditions modes at finite frequencies are related to PEs.) We have therefore averaged all $g_{\alpha PE}(r)$ in the range $0 \le \omega \le \omega_{max}$ in order to get a correlation function that takes into account all the TDs at low frequencies. Since the number of TDs increases quadratically with $\omega$, see Fig. 2d, we have weighted the individual $g_{\alpha PE}$ by $\omega^{-2}$. The resulting correlation functions $g^{av}_{+PE}(r)$ and $g^{av}_{-PE}(r)$ are shown in panel **c** and from this graph one clearly recognizes that the negative TDs are significantly correlated with the PE in that the peak at small $r$ rises above 2.5, which means that there is a more than two-fold increase of the probability that a PE occurs close to a TD as compared to a uniform distribution.

Since these results have been obtained by considering all the modes up to a cut-off frequency $\omega_{max} = 3.5$, it is of interest to check to what extent the correlation between the TDs and PE depends on the frequency of the mode. Therefore we show in Fig. 3d the probability

that a PE is close to a TD as a function of $\omega$. In practice we used a cut-off distance between TD and PE of 1.6, i.e., the location of the first minimum in the radial distribution function between the particles, see Supplementary Fig. 2. The graph shows that the probability is high for small value of $\omega$ and then decreases rapidly if one approaches the threshold 3.5, the value we have used to calculate the curves in panel **(c)**, thus justifying this choice. For larger values of $\omega$ the probability stays at around 0.3, a value that is determined by the spatial density of the PE, i.e., these high frequency modes are not relevant for the occurrence of the PEs.

The results presented so far are for a strain of $\gamma = 0.025$. Also included in panel **d** is the corresponding data for $\gamma = 0.05$, a value that is close to the yielding point of the system, see Fig. 4b below, and one recognizes that also in this case there is a pronounced increase of the probability at small $\omega$, i.e., the negative TDs for small $\omega$ correlate well with the PE and this correlation disappears for $\omega \gtrsim 3.5$, indicating that this value is independent of the strain. Note that this threshold is also close to the frequency at which the number of TDs as a function of frequency starts to show the first deviation from the quadratic law found at low values of $\omega$, see Fig. 1f. Furthermore it also coincides with the frequency at which the density of states starts to show deviations from the Debye regime, see Supplementary Fig. 3b, i.e., above this $\omega$ the nature of the vibrational modes is no longer acoustic and starts to become affected by the disorder of the structure. Therefore one concludes that this threshold does indeed reflect a relevant frequency for the plastic yielding of the sample.

In order to have a visual impression of the correlation between the location of the TDs and the PEs we present in Fig. 3e the charge density field of these TDs (color map) as well as the location of the PEs (symbols). The graph demonstrates the strong correlation between zones with a high density of $-1$ TDs and the PEs. (This strong correlation becomes even more visible if we just consider the $-1$ TDs to generate the colormap, see Supplementary Fig. 7.) One also notices that basically all zones of high density do contain PEs, i.e., the negative TDs are

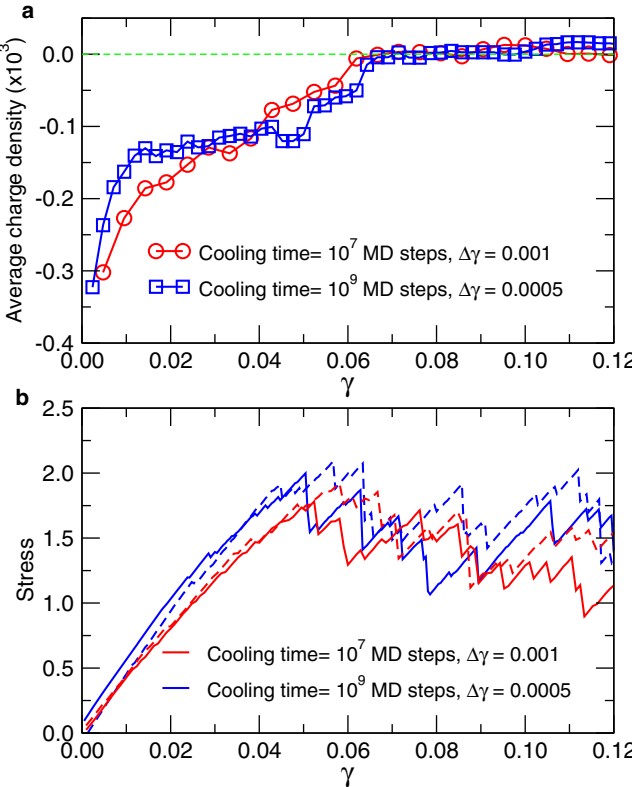

**Fig. 4 | Correlation between plastic events and the topological defects as a function of strain. a** The average charge of the TD, $\overline{C}$, as a function of $\gamma$ is negative for all strains less than the yielding strains. The red and blue curve corresponds to the glass samples that have been quenched with a moderate and slow cooling rate, respectively. **b** Stress-strain curve for the two glass samples. The full and dashed lines correspond to a shear in the positive and negative $x$-direction, respectively.

indeed able to predict the location of the "soft spots", the points at which the sample is yielding. Note that we have done the simple shearing in the $+x$ and the $-x$ direction and, as expected, the location of the corresponding plastic events (filled and open symbols) are not quite the same. However, both types of PEs do show a good correlation with the zones of strong negative charge, i.e., to a first approximation the tensorial character of the plastic response can be neglected. (We also mention that shearing the system in the $y$-direction gives rise to PEs that are spatially strongly correlated with the ones shown in panel (e), indicating that the direction of the shear does not play a very important role in our case, see Supplementary Fig. 8.) One also recognizes clearly the anti-correlation between the PE and the positive TDs (zones in red), showing that the former are forming "hard spots" in the samples that are stable under a shear transformation. This result is reasonable since the positive TDs correspond to the points of the vibrational excitations at which the vibrational amplitude goes smoothly to zero and hence it can be expected that the nearby particles will not be affected strongly by the external shear. This rational is consistent with earlier findings that particles with larger amplitude of the vibrational eigen-vector contribute more to the non-affine deformation and thus irreversible rearrangements[11,12]. Finally, we mention that the good correlation between the position of the negative TD and the plastic events are also found if the former have been obtained at a finite strain, see Supplementary Fig. 9. This shows that our results are robust with respect to minor modifications of the protocol.

The results presented so far correspond to the strain $\gamma = 2.5\%$ and a glass sample that was generated using a moderate cooling rate. To investigate how the correlation between PE and TD depends on the applied strain we identify for a given value of $\gamma$ all particles that are in the top 5% of $D_{min}^2$ and calculate for these particles the value of the

charge density field of the TD, thus giving an average charge $\overline{C}(\gamma)$. This average charge is strongly negative if there is a good correlation between the PE and the $-1$ TD, while $\overline{C} = 0$ if there is no correlation. Figure 4a shows $\overline{C}$ as a function of $\gamma$. For the sample that has been produced with a moderately cooling rate (red curve) the value of $\overline{C}$ increases steadily with increasing $\gamma$, showing that the correlation between the PE and TD is high at small strain and then gradually disappears if the yielding strain is approached, see stress-strain curve in panel (**b**). For the sample that has been well annealed, blue curve, the value of $\overline{C}$ at small $\gamma$ is very close to the one of the first sample, showing that the quality of the correlation does not depend strongly on the nature of the glass if the strain is small. For intermediate strains the value of $\overline{C}$ changes only weakly but once the yielding strain is approached, it rapidly goes to zero, i.e, the correlation is completely lost. Figure 5 displays the maps of the TD charge density field for the two glass samples. The different panels show the location of the PE at the indicated strains, allowing to get a visual perception of the correlation as a function of $\gamma$. Hence Figs. 4 and 5 show that the local TD charge density is an indicator that correlates indeed very well with the location of the PE, independent of the strain or the annealing of the glass.

## Discussion

The identified correlations between the location of the topological defects and the plastic events indicate that these quantities are intimately related to each other. Zones of the sample which have a high density of defects with topological charge $+1$ are stable towards shearing since in their vicinity the amplitude of the vibrational excitations are small and vary smoothly in space, i.e., the application of a local strain will not affect significantly these vibrational modes and thus will not give rise to a plastic event. This is also the case for the boundary between two neighboring regions of $+1$ TD with opposite chirality, since also for this geometry the flow field is a smooth function in space, see Fig. 1e. For zones having a high density of TD with topological charge $-1$, the vector-field of the eigenmodes will instead be strongly modified if one applies a local strain, thus making it highly probable for a plastic event to occur. This high susceptibility can be understood by noting that the geometry of this vector field has similarities to the motion of a particle close to a saddle-point, see Fig. 1e. This makes that a slight perturbation in the potential energy landscape of the system will destabilize the particles in the vicinity of this TD, i.e., lead to a plastic event. We also point out that the geometry of the vector-field close to a $-1$ TD is very similar to the displacement field of a so-called T1 event as discussed by Falk and Langer[9], i.e., the plastic events that are considered to be relevant for the yielding of amorphous systems. This observation is thus further evidence that the TD are indeed relevant indicators for the location of PE. Note that this connection between TD and PE can be expected to hold only if the local shear is indeed a smooth function in space, i.e., if the sample can locally be described as an elastic medium. This is only the case if the length scales considered are sufficiently large, which in turn rationalizes our finding that the correlation between PE and TD is pronounced if the distance between the TDs is larger than the lengths scale on which the system can support acoustic phonons, while for smaller scales the correlation is lost. (This is also coherent with the findings from ref. 13 in which best results were obtained once the size of the probe region were such that continuum mechanics can be applied.) One interesting aspect of our results is thus the fact that the indicator we have introduced here is neither local nor long-ranged since it is related to a weighted average of modes that have different characteristic length scales. This observation might rationalize the finding that some of the indicators studied before, and that were either purely short-ranged or intermediate range, did not give satisfactory predictions for the location of PE[15].

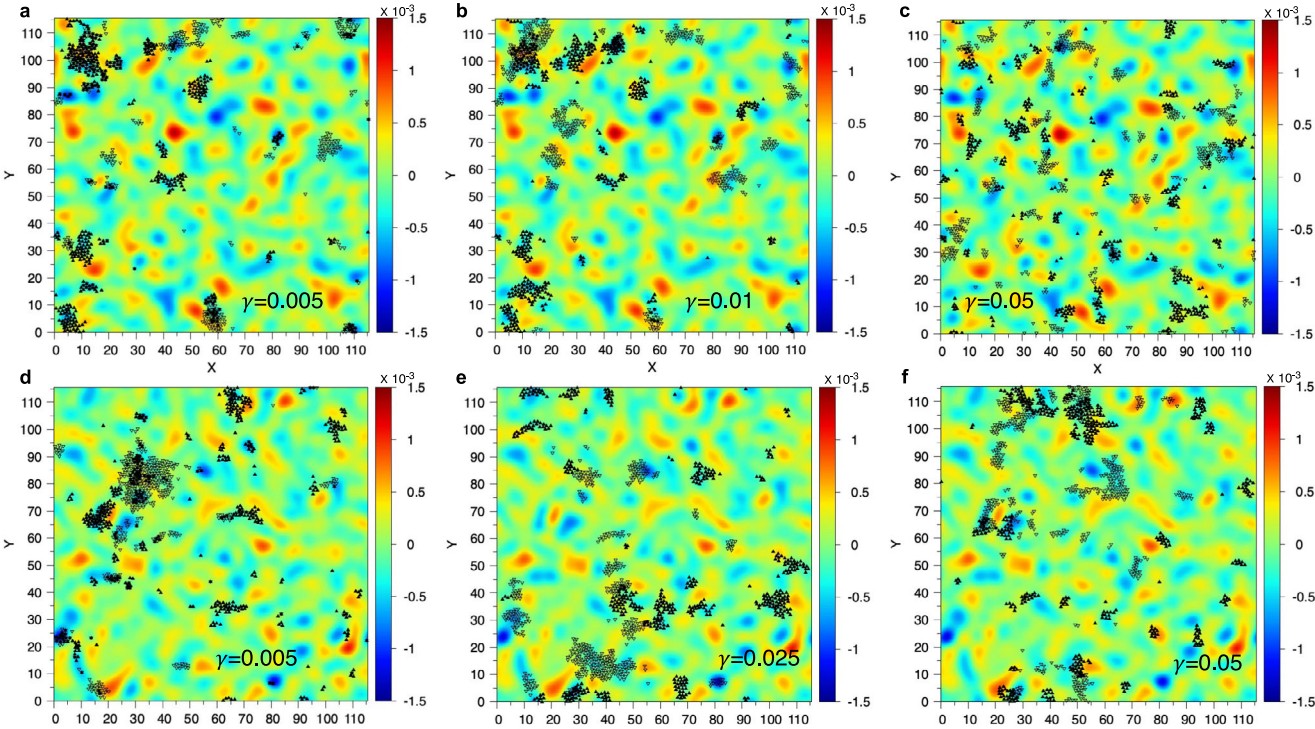

**Fig. 5 | Location of the plastic events and map of the TD charge field.** Maps of the charge field of the TD, including the PE at different values of strain (stated in the panels). The filled and open symbols show the location of the plastic events when the sample is sheared in the positive and negative $x$-direction, respectively. Panels **(a–c)** are for the sample quenched to the glass state within $10^7$ time steps and **(d–f)** for the sample that was cooled more slowly ($10^9$ time steps).

Our results also shed light on the previous studies in which machine-learning approaches have been used to predict the location of PE[16,40–43,46]. Despite the success of these approaches, the understanding on the physical origin of these soft spots has so far not really been elucidated and our findings do now allow us to interpret these zones in terms of a simple physical quantity. Note that in contrast to previous studies, refs. 30,31,38,39, we do not have to determine the normal modes as a function of the strain in order to predict the location of the PE. Instead this location is obtained directly from the normal modes at strain zero, i.e., it is a purely structural quantity in the unperturbed system. This feature should thus permit to make headway with analytical calculations that aim to describe the yielding of amorphous systems and hence to obtain a better understanding of this important problem. Furthermore it will also be interesting to probe how the topology of the vibrational modes at strain zero investigated here are related to the topological defects of the displacement field at finite strain, since the latter have been found to correlate well with the location of the plastic events[39]. The evolution of the vibrational modes with strain should thus allow to gain a deeper understanding of yielding, and notably on the difference between ductile and brittle fracture. While in the present work we have focused on the properties of the atoms, some recent studies on amorphous systems have suggested that also the electronic degrees of freedom show some interesting features related to topological defects of the wave-functions[47]. How these defects are related to the ones investigated here is an open question which should be addressed in future studies. Finally, we mention that the approach presented here can in principle be generalized in a straightforward manner to three-dimensional systems since also in this case the eigen-vector field can be computed easily. Subsequently one can determine the locations of the topological defects, which in this case will be lines[48], and see whether they correlate with the position of the PE

of the sheared sample. This approach might thus be an interesting alternative to other methods that have been proposed to predict the location of PE in 3D[16,49].

## Methods

### MD simulation

The two-dimensional glass former we study is an equimolar binary mixture of particles with size $\sigma_1$ and $\sigma_2$ that interact via a truncated Lennard-Jones potential:

$$u_{ab}(r) = 4\varepsilon \left[ \left( \frac{\sigma_{ab}}{r} \right)^{12} - \left( \frac{\sigma_{ab}}{r} \right)^6 \right] + C_{ab}. \tag{1}$$

Here $\sigma_{12} = (\sigma_1 + \sigma_2)/2$ and the constant $C_{ab}$ ensures that $u_{ab} = 0$ at $r_{cut} = 2^{1/6}\sigma_{ab}$, so the potential is purely repulsive and continuous at the cutoff distance. The size ratio was set to 1.414 to prevent crystallization[50,51]. A total of $N = 10000$ (or $N = 1250$) particles with the mass ratio $m_1/m_2 = (\sigma_1/\sigma_2)^2$ were enclosed in a square box of length $L = 115.47$ (number density is 0.75) with periodic boundary conditions. The units of length, mass, and energy are $\sigma_1$, $m_1$, and $\varepsilon$, respectively. Time and temperature are in units of $\tau = \sigma_1 \sqrt{m_1/\varepsilon}$ and $\varepsilon/k_B$, with $k_B$ the Boltzmann constant. The time step of integration was $0.001\tau$. We first equilibrated the system in the liquid state at $T = 5.0$ during $10^6$ MD steps and then quenched it to the final temperature $T = 0.1$ within $10^7$ time steps using a linear cooling schedule, after which we annealed the sample for another $10^6$ MD steps. The Supplementary Fig. 5 shows that for this system the mode-coupling temperature is around $T_{MCT} = 1.09$, i.e., our initial temperature is about five times higher. The cooling rate used to produce our sample makes that the glass-former falls out of equilibrium at around $T_{MCT}$, i.e., at a temperature at which the dynamics is already quite sluggish. In order to make sure that the results presented for this sample are not affected in a significant manner by this cooling rate, we have repeated the analysis for a

sample that has been cooled 100 times slower ($10^9$ time steps for the quench).

## Normal modes
The zero-temperature glass was generated by a conjugate gradient energy minimization process, and the vibrational normal modes were obtained by diagonalizing the dynamical matrix.

## Topological defects
To identify the topological defects of an eigenvector field $(e_i^x, e_i^y)$, $i = 1, \ldots 2N$, we first assigned an angle $\theta(\vec{r})$ on every site $\vec{r}$ of a $100 \times 100$ square lattice superposed to the sample. (We have checked that a $120 \times 120$ grid does not change the main conclusions presented in the main text.) This angle $\theta(\vec{r})$ is defined to be

$$\tan\theta\left(\vec{r}\right) = \sum_i w\left(\vec{r} - \vec{r_i}\right)e_i^y / \sum_i w\left(\vec{r} - \vec{r_i}\right)e_i^x, \quad (2)$$

where $\vec{r_i}$ is the location of particle $i$, and $w(\vec{r} - \vec{r_i})$ is a weight function (in practice a Gaussian[52,53]: $w(\vec{r} - \vec{r_i}) = \exp(-|\vec{r} - \vec{r_i}|^2/\sigma^2)$ with $\sigma = 1$). Hence this procedure allows to define a map from the eigenvector field, given at the positions of the particles, to a regular grid. The topological defects were then identified by calculating the line integral of $\nabla\theta$ over a closed path on the lattice giving $0$, $2\pi$, or $-2\pi$ if the loop contained no, a $+1$ TD, or a $-1$ TD, respectively. The location of a TD was obtained by identifying the center of the smallest square with a non-zero value of the line integral.

## Plastic events
The athermal and quasistatic shear was realized by shearing the simulation box by a small strain increment $\Delta\gamma$ and subsequently minimizing the energy of the configuration with a conjugate gradient algorithm. This procedure was repeated until the global strain reached $\gamma = 2.5\%$, using $\Delta\gamma = 0.05\%$, (and $5\%$, using $\Delta\gamma = 0.1\%$), the strain at which the data in Fig. 3 is shown.

The locally irreversible rearrangements of particles in the glass are evaluated by the non-affine displacement $D_{min}^2$ introduced in ref. 9 and which is defined as the minimum of

$$D_i^2 = \frac{1}{N_i}\sum_{j(i)}\left[\vec{r}_j(\gamma) - \vec{r}_i(\gamma) - \mathbf{F}_i \times (\vec{r}_j(\gamma - \Delta\gamma) - \vec{r}_i(\gamma - \Delta\gamma))\right]^2, \quad (3)$$

where the index $j(i)$ runs over all the particles that are nearest neighbors of particle $i$ (thus the distance is less than 1.6, the first minimum of the total pair correlation function), $\vec{r}_j(\gamma)$ denotes the position of particle $j$ after the minimization of the box that is sheared by $\gamma$, and $N_i$ is the number of nearest neighbors of particle $i$. The matrix $\mathbf{F}_i$ is chosen such that it minimizes $D_i^2$ and it can be calculated based on the spatial coordinates of the corresponding particles[9].

## Correlation function between the TDs and the PEs
For each eigenmode $\kappa = 1, 2, \ldots, 2N$, we define the radial pair correlation function $g_{\kappa,\alpha\text{PE}}(r)$ between the TDs and PEs, with $\alpha \in \{-1, +1\}$, as

$$g_{\kappa,\alpha\text{PE}}(r) = \frac{L^2}{2\pi r N_{\text{TD}} N_{\text{PE}}}\sum_{i=1}^{N_{\text{TD}}}\sum_{j=1}^{N_{\text{PE}}}\delta\left(r - |\vec{r}_{ij}|\right). \quad (4)$$

Here $N_{\text{TD}}$ and $N_{\text{PE}}$ are the number of TD of the mode $\kappa$ and the number of particles associated to PEs, respectively, and $r_{ij}$ is the distance between the TD $i$ and a particle in the PE $j$. The average

correlation function $g_{\alpha\text{PE}}(r)$ is then given by

$$g_{\alpha\text{PE}}(r) = \frac{\sum_\kappa g_{\kappa,\alpha\text{PE}}(r)/\omega_\kappa^2}{\sum_\kappa 1/\omega_\kappa^2}, \quad (5)$$

where the sum over $\kappa$ runs up to the cut-off frequency $\omega_{\max}$ defined in the main text.

## Data availability
The datasets generated during and/or analysed during the current study are available from the corresponding author on reasonable request.

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

## Acknowledgements

We thank Z.Y. Huang, M.Z. Li, P.F. Guan, and the other members of the "Beijing Metallic Glass Club" for the long-term fruitful discussions and E. Bouchbinder, E. Lerner, D. Richard, S. Patinet, D. Vandembroucq and A. Zaccone for useful comments on the manuscript. W.K. is a senior member of the Institut universitaire de France. This work was supported by the National Natural Science Foundation of China (Grant Nos. 11804027, 11935002, and 52031016), and the National Key R&D Program (No. 2021YFA1400500).

## Author contributions

Z.W.W., W.K., and L.X. designed the project with advice from W.H.W. Z.W.W. carried out the simulations, and Y.C. assisted in the identification and analysis of the TDs. Z.W.W., W.K., L.X., and W.H.W analyzed the data. Z.W.W., W.K., and L.X. wrote the paper.

## Competing interests

The authors declare no competing interests.
