## [Peer Review File · Nature Communications]

Topology of vibrational modes predicts plastic events in glassesREVIEWER COMMENTS

Reviewer #1 (Remarks to the Author):

The report is attached below.

Reviewer #2 (Remarks to the Author):

The paper "Topology of vibrational modes predicts plastic events in glasses" describes an investigation wherein the authors test the hypothesis that the plastic susceptibility of a 2D simulated purely-repulsive amorphous system can be predicted by measuring the topological defects in its vibrational eigenmodes. The idea is interesting and novel, and the paper provides valuable insights. It seems intuitive that the negative topological defects would correspond to shear-transformation zones (STZs) since these singularities have displacement fields that corresponds closely with observed STZ rearrangements (at their simplest, atomistic T1 events) and share the symmetry of the quadrupolar stress fields associated with such activity. I believe that this work is worthy of publication if some of the loose ends the authors do not adequately discuss can be addressed. Below are my questions:

1. The system that the authors choose to simulate is 2D and purely repulsive. The sample is prepared by a quench that proceeds from $T=5$ (approx 15 times the glass transition temperature) to $T=0.1$ (about 1/3 of T_g) in about 10,000 LJ time units. To give some physical interpretation of this time-scale, for an atomistic system an LJ time unit is about a picosecond and a typical T_g might be around 600K. So, this corresponds roughly to quenching through T_g at about 300K/ns. Even by MD standards this is very high. So, these systems are in the extremely rapidly quenched (ERQ) regime. This is significant because in Ref. [15] Richard, et al. showed that predictors that are highly predictive for identifying plastic activity in ERQ glasses are not as highly predictive in deeply quenched glasses. It is particularly notable that the predictors that relied on analysis of normal modes did well in the ERQ glasses, but fared significantly less well in more stable amorphous systems, particularly in the ultra-stable regime.

EITHER The authors should be clear in their exposition about the nature of their glasses being extremely rapidly quenched. They should discuss the regime they correspond to vis-a-vis Ref. 15 and explain the limitations this implies for prediction outside the regime of ERQ glasses.

OR The authors should do further studies on a broader range of amorphous systems including ultra-stable system and include theses in the study so that the reader can be more confident that this correspondence between plastic activity and topological defects in the vibrational spectrum is not limited to the ERQ regime.

2. The authors provide evidence of good correlations between this metric and locations of plastic activity as determined via the D_{2min} measure. However, the way that this correlation is reported is significantly different from Ref. 15 which compared a large number of other, possibly comparable metrics. Using some of the same means of analysis, such as reporting the correlation decay as was shown in Fig 6 of that paper would make these results comparable and increase the value of this paper greatly. It seems that it would not take much additional work.

3. The authors make the claim that "Finally we mention that the approached (sic) presented here can be generalized in a straightforward manner to three dimensional systems and it will thus be highly interesting to determine for these cases the correlation between PE and TD."

This is puzzling to the reader because topological defects can differ in significant ways in 3D from 2D. See for example (Sajedah Afghah, Robin L. B. Selinger & Jonathan V. Selinger (2018) Visualising the crossover between 3D and 2D topological defects in nematic liquid crystals, *Liquid Crystals*, 45:13-15, 2022-2032, DOI: 10.1080/02678292.2018.1494857). There have been recent advances measuring local yield stress in 3D (<https://doi.org/10.1016/j.jmps.2021.104671>). So, describing the potential connection between these approaches would be highly valuable. However, some significant further elucidation is necessary because the generalization and connection to 3D LYS and STZ structure is far from obvious, at least to this reader.

4. The description of how the topological defects are measured in the methods section on pages 10-11 seems unclear and possibly incorrect. The authors say, "The topological defects were then identified by integrating θ over a closed path on the lattice, and located by taking the position of the center of the smaller square with a value of 2π (-2π) of this integral, defining thus charges of +1 and -1 respectively. I believe that (a) this is not quite correct in that the authors are integrating $d\theta$, i.e. the derivative of θ w.r.t. the contour s parameterized on the interval from $[0, 1]$ with respect to ds over the contour length, and (b) they should specify that this is being done in a counter-clockwise fashion otherwise the signs are reversed. Description at this level of detail also does seem potentially relevant to the question of 3D/2D discussed above in point 3.

Reviewer #3 (Remarks to the Author):

The plastic flow of amorphous systems consists of series of localized rearrangements of the disordered structure (plastic events). The understanding of the relation between the plastic behavior and the amorphous structure remains elusive. In particular it is not clear to what extent it is possible to adapt the notion of defect to amorphous systems and to identify specific patterns of the disordered structure associated with plastic events.

In the context of this very active field of research, the authors of the present manuscript develop an original approach: they adapt a topological description early used to characterize (real space) structural defects to the displacement field associated to low frequency vibrational modes.

In particular, the authors show convincingly the correlation between the location of negatively charged topological defects of the vibrational modes of a 2D model glass and the location of plastic events obtained by shearing (backward and forward) the same amorphous system.

The manuscript is very well written. The results are novel, interesting and stimulating. I think that the manuscript can be published in *Nature Communications* provided the authors consider the following points.

The approach that the authors develop for the vibrational modes is reminiscent of early attempts of identifying topological defects in the real space amorphous structure (disclination, frustration, etc.). This is very briefly mentioned in the abstract/introduction but the authors may want to emphasize and discuss this point in a more explicit way in their manuscript.

A limit of the present study is that the tensorial character of the plastic behavior is not fully taken into account. Depending on the direction chosen to shear the system, different plastic events can actually be unveiled. When shearing along only one direction (even backward and forward) one gets only a subset of the potential plastic events. I think the authors should at least use the other shear direction. This would allow them to reveal if not all, at least a

significant additional population of plastic events and this will likely improve the level of correlation they get between topological defects and plastic events.

Damien Vandembroucq

The paper by Wu and collaborators discuss the possibility of predicting plasticity in glasses using topological properties of the vibrational modes. More precisely, they analyze the topological properties of the eigenvector field as a function of the vibrational frequency. The main finding of this work is the discovery of a correlation between topological defects in the eigenvector field with negative charge and the soft spots responsible for plasticity. This provides a direct link between the topological properties of the structure before the deformation and the plastic behavior during it.

I find the analysis robust and the results interesting and promising. I do have some concerns about the physical interpretation of the results which I believe should be addressed before I can recommend this manuscript for publication in Nature Communications.

Below, I list my major and minor concerns (in random order). I encourage the authors to consider them carefully and improve their manuscript accordingly.

1. There is an increasing interest regarding topology in amorphous systems and glasses. Here, the authors focus on the problem of defects and plasticity but there are also recent attempts to describe the glass transition and the anomalous properties of glasses using topological defects (arXiv:1902.06593, arXiv:2204.08737). I believe it would be nice to add a short paragraph in the introduction and discuss more broadly the connection between topology and amorphous systems. See also arXiv:2010.02851 for example.
2. More importantly, recently, Ref.[39] used as well topological objects and their properties to predict plastic events and yielding in amorphous systems (see also arXiv:2101.05015 for a companion paper which should also probably be mentioned). The approach therein seems very related to the idea of this manuscript but also present some fundamental differences. In particular, if I understand it correctly, the main difference is that in Ref.[39] the authors obtain the topological defects from the displacement vector field, which is a quantity defined under deformation, while in this manuscript the authors are able to make a step forward and to identify the defects directly from the structure prior deformation. Is this correct? Given that the philosophy and central idea of the two works is very close, and that Ref.[39] is a direct predecessor of this paper, I believe that a more in depth discussion about the similarities and differences with Ref.[39] in the conclusions and in the introduction is in order.
3. In Fig.1(f), after the initial scaling regime $N \sim \omega^2$, there seems to be a second regime with a faster scaling. The first region coincides with the Debye regime where the DOS is quadratic and the dispersion of the modes linear (see explanation around lines 187-190). It would be interesting to understand better the relation between the scaling regions in the number of defects and the features in the DOS. For example, if the DOS is normalized by the Debye value, is the BP frequency correlated with

the crossover frequency between the two scaling regimes in the number of defects discussed above? It seems so. Is this second faster scaling a consequence of the quasi-localised modes emerging at larger frequency on top of acoustic modes? More in general, is there any property of the topological defects which connects to the BP anomaly? As a curiosity, it would be interesting to see the number of defects and the DOS, both normalized by ω^2 , as a function of the frequency ω . That might reveal potential connections between the two.

4. Continuing in this direction, around lines 178-179, the authors make a very interesting statement that modes with frequency above a certain cutoff are not relevant for PE. Even more, from the discussion about figure 1(f) and that around lines 187-190, such a cutoff frequency corresponds indeed with the edge of the region in which the number of defects scales like ω^2 and the vibrational modes are of acoustic nature, $\omega \sim q$. Then, I find this remarkable since it suggests that plastic events are related to Debye-like acoustic modes and not to quasi-localised modes which become important only at higher frequency (cfr. boson peak, etc). I suggest the authors to expand on these lines and explicitly state whether their results can support any conclusions regarding the nature of the vibrational modes responsible for plasticity or at least correlated with PE.
5. Related to the previous point, in principle, for rigorous athermal quasi-static shear (AQS), we expect that only the lowest mode (or modes), and therefore the negative defects therein, are activated. Nevertheless, Fig. 3b suggests that negative defects beyond the lowest frequency modes can also contribute to plasticity. The “activation energy” of the defects is not discussed. I understand that this might be a highly non-trivial question which goes beyond the scope of the current paper but it would be nice to comment on this at least.
6. Also, in lines 202-203, the authors claim that the tensorial nature of the plastic response can be neglected at leading order. Naively, this seems at odd with the proposal about the fundamental role of quadrupole and dipole structures for plasticity and shear zones in amorphous materials. A comment about this point would be helpful to put the results in a broader context.
7. In lines 123-124, the authors say that the correlation functions for TD with charge +1 resemble those of liquids. Nevertheless, it seems to me that all of them are almost featureless after the first peak. This situation seems closer to that of gases, or eventually very dilute liquids. Also, this suggests that only two-body correlations are important, while three-body and even higher-order correlations are all essentially weak. This is not the case in liquids, where many-body correlations are significant. I would like the authors to comment on this point and, if they agree with me,

to modify accordingly that statement. Moreover, it would be interesting to think further about what this is telling us about the interactions and dynamics of the positive TD.

8. In Fig.1(e) and discussion surrounding it, the authors mention the chirality of the positive defects. I wonder if the chirality plays also an important role somewhere. In addition, can the chirality be defined also for negative TD? It would be interesting to think along these lines and comment on this.
9. If I understand correctly the analysis, the plastic events are calculated at finite strain intervals while the features of the TD from the undeformed structure. I wonder how the analysis would change if one were to consider the structure and the TD therein at a nonzero strain step, for example the one before the plastic events occur. I do not expect the overall conclusions to change but it would be interesting to give it a try and compare the two scenarios.
10. Last but not least, I find the observations of the authors very interesting but I miss any physical explanations for them. In particular, is there any heuristic argument why only topological defects with negative charge should correlate with plasticity? Which is the major physical difference between defects with negative and positive charge? Why defects with positive charge should be irrelevant for plasticity? A discussion of the results in terms of (even speculative) physical arguments is needed in my opinion. Otherwise, the results are left as a mere empirical observation from simulations.

When these issues are solved, I will be happy to suggest the publication of this work in Nature Communications.

Response to the Reviewer's Comments for NCOMMS-22-45560-T

Zhen Wei Wu, Yixiao Chen, Wei-Hua Wang, Walter Kob, and Limei Xu

Response to Reviewer #1

The paper by Wu and collaborators discuss the possibility of predicting plasticity in glasses using topological properties of the vibrational modes. More precisely, they analyze the topological properties of the eigenvector field as a function of the vibrational frequency. The main finding of this work is the discovery of a correlation between topological defects in the eigenvector field with negative charge and the soft spots responsible for plasticity. This provides a direct link between the topological properties of the structure before the deformation and the plastic behavior during it.

I find the analysis robust and the results interesting and promising. I do have some concerns about the physical interpretation of the results which I believe should be addressed before I can recommend this manuscript for publication in Nature Communications.

Below, I list my major and minor concerns (in random order). I encourage the authors to consider them carefully and improve their manuscript accordingly.

We are happy to read that the reviewer acknowledges our analysis to be robust and the results interesting and promising.

1. There is an increasing interest regarding topology in amorphous systems and glasses. Here, the authors focus on the problem of defects and plasticity but there are also recent attempts to describe the glass transition and the anomalous properties of glasses using topological defects (arXiv:1902.06593, arXiv:2204.08737). I believe it would be nice to add a short paragraph in the introduction and discuss more broadly the connection between topology and amorphous systems. See also arXiv:2010.02851 for example.

We thank the reviewer for this suggestion and for having pointed out these references to us. (A similar remark has also been made by Ref. #3.) In the revised manuscript, we have added in the introduction the following text in which we mentioned two of the three papers pointed out by the referee. The paper arXiv:2010.02851 is about the topological phases of electrons in amorphous systems and is thus not directly related to the present study and therefore we do not mention it in the Introduction. However, we do now refer to this work, or rather to a very recent paper by this author on the same topic Ref.[6,47], in the Discussion:

Line 74:

...topological properties. The idea that topological properties might be useful to understand certain thermodynamic and kinetic features of glasses has been suggested only in recent years [44,45]. Subsequently Baggioli et al. have put forward the relevance of topological properties by showing that plasticity is mediated by topological features in the non-affine displacement field of glasses under deformation [39]. In the present work we find that...

New text in Discussion:

While in the present work we have focused on the properties of the atoms, some recent studies on amorphous systems have suggested that also the electronic degrees of freedom show some interesting features related to topological defects of the wave-functions [47]. How these defects are related to the ones investigated here is an open question which should be addressed in future studies.

2. More importantly, recently, Ref.[39] used as well topological objects and their properties to predict plastic events and yielding in amorphous systems (see also arXiv:2101.05015 for a companion paper which should also probably be mentioned). The approach therein seems very related to the idea of this manuscript but also present some fundamental differences. In particular, if I understand it correctly, the main difference is that in Ref.[39] the authors obtain the topological defects from the displacement vector field, which is a quantity defined under deformation, while in this manuscript the authors are able to make a step forward and to identify the defects directly from the structure prior deformation. Is this correct? Given that the philosophy and central idea of the two works is very close, and that Ref.[39] is a direct predecessor of this paper, I believe that a more in depth discussion about the similarities and differences with Ref.[39] in the conclusions and in the introduction is in order.

We thank the referee for this comment. We completely agree with this description regarding the difference between Ref. [39] and our work. We have now added a short discussion on this point in the summary in the Discussion section of the manuscript:

Line 292:

...understanding of this important problem. Furthermore it will also be interesting to probe how the topology of the vibrational modes at strain zero investigated here are related to the topological defects of the displacement field at finite strain, since the latter have been found to correlate well with the location of the plastic events, Ref. [39]. The evolution of the vibrational modes with strain should thus allow to gain a deeper understanding of yielding, and notably on the difference between ductile and brittle fracture. While in the present work we have focused on the properties of the atoms, some recent studies on amorphous systems have suggested that also the electronic degrees of freedom show some interesting features related to topological defects of the wave-functions [47]. How these

defects are related to the ones investigated here is an open question which should be addressed in future studies.

3. In Fig.1(f), after the initial scaling regime $N \sim \omega^2$, there seems to be a second regime with a faster scaling. The first region coincides with the Debye regime where the DOS is quadratic and the dispersion of the modes linear (see explanation around lines 187-190). It would be interesting to understand better the relation between the scaling regions in the number of defects and the features in the DOS. For example, if the DOS is normalized by the Debye value, is the BP frequency correlated with the crossover frequency between the two scaling regimes in the number of defects discussed above? It seems so. Is this second faster scaling a consequence of the quasi-localised modes emerging at larger frequency on top of acoustic modes? More in general, is there any property of the topological defects which connects to the BP anomaly? As a curiosity, it would be interesting to see the number of defects and the DOS, both normalized by ω^2 , as a function of the frequency ω . That might reveal potential connections between the two.

We are a bit puzzled that the referee uses the expression “faster scaling”. In Fig.1f the increase of the number of topological defects is *less* steep for $\omega > 3$ than the increase at lower frequencies. So we would describe this as a “slower scaling”. Anyway, the main point the referee raises concerns the connection between topological defects and the BP anomaly. This is indeed highly interesting question and has been the subject of a large number of previous studies. However, since the subject of the present paper are the origin of the plastic events, the question regarding the boson peak is well beyond the scope of the present manuscript but we plan to address this question in more detail in a future study. Already here we point out, however, that, as suggested by the referee, there seems to be indeed a connection between the end of the Debye regime in the density of states and the crossover in the ω^2 -dependence in the number of topological effects. Such a relation does make sense, since in the manuscript we argue that the ω^2 -dependence of the number of topological defects, Fig. 1(f), is related to the fact that the system can be described as an elastic solid. This approximation breaks down at the frequency at which the Debye description is no longer valid, thus rationalizing that these two crossover points are very close to each other. This result is now documented in Extended Data Fig. 3b where we show the density of states as well as the number of defects (also reproduced here as Fig. R1): Both curves start to drop at the same frequency, indicating that the two crossover frequencies are indeed related to each other. To discuss this result we have added the following text:

Line 119:

At $\omega \approx 3.5$ this quadratic dependence crosses over to a weaker one, most likely because the glass can no longer be considered as an elastic medium. This observation is in harmony with the fact that at this frequency also the density of states ceases to show the linear dependence on ω expected from the Debye-law, see Extended Data Fig. 3(b). This coincidence also hints a possible relation between the TD and the so-called boson peak [6], i.e., this peak might be related to the fact that on length scales smaller than the corresponding

acoustic modes the notion of homogeneous elasticity is no longer applicable. The details of this possible connection should be investigated in future work.

Figure R1: The number of topological defects per degree of freedom ($2N$), divided by ω^2 , and the vibrational density of states, divided by ω , as a function of ω .

4. Continuing in this direction, around lines 178-179, the authors make a very interesting statement that modes with frequency above a certain cutoff are not relevant for PE. Even more, from the discussion about figure 1(f) and that around lines 187-190, such a cutoff frequency corresponds indeed with the edge of the region in which the number of defects scales like ω^2 and the vibrational modes are of acoustic nature, $\omega \sim q$. Then, I find this remarkable since it suggests that plastic events are related to Debye-like acoustic modes and not to quasi-localised modes which become important only at higher frequency (cfr. boson peak, etc). I suggest the authors to expand on these lines and explicitly state whether their results can support any conclusions regarding the nature of the vibrational modes responsible for plasticity or at least correlated with PE.

We thank the referee for pointing out this interesting connection. This observation is closely related to the point #3 of the referee. Thus please refer to our answer and the text we added to the manuscript in point #3.

5. Related to the previous point, in principle, for rigorous athermal quasi-static shear (AQS), we expect that only the lowest mode (or modes), and therefore the negative defects therein, are activated. Nevertheless, Fig. 3b suggests that negative defects beyond the lowest frequency modes can also contribute to plasticity. The “activation energy” of

the defects is not discussed. I understand that this might be a highly non-trivial question which goes beyond the scope of the current paper but it would be nice to comment on this at least.

We agree with the observation made by the referee that there is indeed a correlation between the plastic events and the topological defects of modes that have not really a low frequency (say up to $\omega = 3.0$), and that this is somewhat surprising. However, we do not think that under quasi-static shear conditions “only the lowest mode” is relevant. The frequency of the lowest mode is determined by the size of the sample ($=L$) in that it decreases like L^{-1} . However, one expects that the location of the plastic events is not strongly dependent on L (if L is reasonably large), since the mechanical softness of a region containing a plastic event is a *local* property. Therefore we expect that the PE are not only influenced by the modes with the lowest frequencies, but by all modes *up to a certain frequency* and Fig. 3b shows that this threshold is around $\omega = 3$. This result is thus in harmony with the fact that above this threshold the acoustic modes start to become strongly scattered by the disorder (as pointed out by the referee) and hence a simple elastic description of the system is no longer possible (as already written in the original version of the text). This conclusion also agrees with the remarks #3 and #4 of the referee that this threshold is likely to have a connection with the boson peak.

Regarding the comment of the referee on the activation energy: This is indeed an interesting question, but, as also remarked by the referee, it is beyond the scope of the paper. However, we hope to address this point in future studies.

We have added the following text to emphasize the point on the low frequencies:

Line 173:

... there is no correlation. (Note that in panel (b) we find a non-trivial correlation even for frequencies that are not small, e.g., $\omega = 1.5$, which indicates that even under quasistatic shear conditions modes at finite frequencies are related to PE's.)

6. Also, in lines 202-203, the authors claim that the tensorial nature of the plastic response can be neglected at leading order. Naively, this seems at odd with the proposal about the fundamental role of quadrupole and dipole structures for plasticity and shear zones in amorphous materials. A comment about this point would be helpful to put the results in a broader context.

We thank the referee for this interesting remark. The quadrupolar structure for the displacement field is an exact result for an elastic media that includes one defect. Previous numerical studies have shown that glasses do indeed show this behavior (Ref.[30] and [38]). However, the question we address in our manuscript is different, since we are neither probing the displacement field, nor are we looking at the limit of one isolated plastic defect (our sample has quite a few and they are not isolated). In view of the relatively high density of the PE present in our sample, it is unlikely that there is a remnant of a quadrupolar

structure of the displacement field, since such a structure is expected to be washed out. Therefore we cannot make any statement on this point. However, in the case of very small strain, i.e., when the density of plastic events is very small, there might indeed be a correlation between the position of PE due to the presence of the long-ranged displacement field. Although we plan to investigate this point in the future using significantly larger system sizes, this question is presently not answerable with the system size we currently use.

7. In lines 123-124, the authors say that the correlation functions for TD with charge +1 resemble those of liquids. Nevertheless, it seems to me that all of them are almost featureless after the first peak. This situation seems closer to that of gases, or eventually very dilute liquids. Also, this suggests that only two-body correlations are important, while three-body and even higher-order correlations are all essentially weak. This is not the case in liquids, where many-body correlations are significant. I would like the authors to comment on this point and, if they agree with me, to modify accordingly that statement. Moreover, it would be interesting to think further about what this is telling us about the interactions and dynamics of the positive TD.

We thank the referee for bringing up this point. We agree that the peaks in Fig. 2a are not very high and that the second, and even more the third peaks, are quite small. However, for a gas one would not observe neither a first peak nor a second peak since by definition of a gas the particles are not correlated. Therefore the structure we have is closer to a liquid than to a gas, and the snapshots from Fig. 1a-c confirm this. On the other hand we agree with the referee that the height of the peaks in the radial distribution functions are not large and therefore the nearest neighbor correlations is indeed quite weak. Thus it is indeed appropriate to describe the structure to be similar to the one of a *hot* liquid (which has no pronounced long range structure but detectable short range structure).

Regarding the presence of many-body correlations: First of all one should recall that in the theory of liquids already simple two-body approximations, like the well-known Percus-Yevick approximation, give rise to a radial distribution function that describes very well the structure of liquids at their triple point. This approximation does not take into account triple (or higher) correlations, which demonstrates that these correlations are not really needed to get a *reasonable* description of the structure. In other words, from the $g(r)$ alone it is hard to tell whether or not higher order correlation functions are relevant. To answer this question one would have to make a serious analysis of the structure, i.e., calculate the three-body correlation function $c_3(\mathbf{r}_1, \mathbf{r}_2)$, which is the excess of the three-body correlation over the one expected from the convolution approximation that involves the two-body correlations (see, e.g, the textbook by Hansen and McDonald on liquid state theory). However, from our own experience to calculate $c_3(\mathbf{r}_1, \mathbf{r}_2)$ (for a different system) it is quite difficult to obtain this function with a good precision and the data we have now for the “TD liquid” is certainly insufficient to do such a calculation.

In order to clarify the nature of the structure we have now modified the text on this point:

Line 136:

... form a structure that is reminiscent of the one of a hot liquid ...

8. In Fig.1(e) and discussion surrounding it, the authors mention the chirality of the positive defects. I wonder if the chirality plays also an important role somewhere. In addition, can the chirality be defined also for negative TD? It would be interesting to think along these lines and comment on this.

This is an interesting point. To our best knowledge, there is no well-defined chirality for negative TD's. This has the interesting consequence that positive TD can be considered to have more structure than negative TD since their chirality allows to divide them into two sub-sets (positive rotation and negative rotation). In other words, positive TD have less symmetry than the negative TD. This is the reason why one sees in the snapshots, Fig. 1, that the positive TD are more ordered than the negative ones. Having said this, we are not aware of any direct implication of this fact for the physical quantities we considered, but it is certainly something that one should think about in the future.

9. If I understand correctly the analysis, the plastic events are calculated at finite strain intervals while the features of the TD from the undeformed structure. I wonder how the analysis would change if one were to consider the structure and the TD therein at a nonzero strain step, for example the one before the plastic events occur. I do not expect the overall conclusions to change but it would be interesting to give it a try and compare the two scenarios.

Figure R2: (a) The stress-strain curve with a mark at the strain at which the charge density map shown in panel (b) was constructed. (b) The charge density map and the associated PE.

The referee is right in that in this work we are aiming to predict the potential plastic events under simple shear using the *undeformed* glass structure. In fact the question raised by the referee is very interesting. In order to answer it we have tested our approach for the system that has $N = 1250$ particles. We have constructed the corresponding charge density map, which predicts the location of the plastic events, by using the particle configuration just before a drop in the stress-strain curve, see Fig. R2(a), and superimposed the associated PE to this map. The result is shown in Fig. R2(b). In agreement with the expectation of the referee (and ours), one sees that also for this case there is a good correlation between the PE and the spots at which the charge density is very negative. Since this indeed an important point and insight, we have added Fig. R2 to the Extended Data and mention this in the main text:

Line 228:

Finally we mention that the good correlation between the position of the negative TD and the plastic events are also found if the former have been obtained at a finite strain, see Extended Data Fig. 9. This shows that our results are robust with respect to minor modifications of the protocol.

10. Last but not least, I find the observations of the authors very interesting but I miss any physical explanations for them. In particular, is there any heuristic argument why only topological defects with negative charge should correlate with plasticity? Which is the major physical difference between defects with negative and positive charge? Why defects with positive charge should be irrelevant for plasticity? A discussion of the results in terms of (even speculative) physical arguments is needed in my opinion. Otherwise, the results are left as a mere empirical observation from simulations.

We thank the referee for this comment since it shows that the physical arguments we put forward in the text to rationalize our findings were not sufficiently clear. (We mention, however, that Referee #2 seems to have gotten the point.) The TD with positive charge are stable configurations since a small disturbance does not alter the local geometry of the eigenmodes. In contrast to this, TD with negative charge are associated with a flow field of the eigenvector which corresponds to a saddle point (see Fig. 1e). This has the consequence that the configuration is locally unstable since a small external perturbation (due to shear) will change the (local) direction of the flow field and thus destabilize the delicate balance of the forces on the particles. We also point out that the motion suggested by the flow field at a TD with negative charge corresponds to the so-called T1-events introduced by Falk and Langer, Ref. [9], and which have been shown to be the most relevant excitations for plastic events in sheared amorphous systems. Thus our finding is indeed compatible with this view. In order to clarify this point we have modified the text on this in the Discussion which now reads:

Line 254:

Zones of the sample which have a high density of defects with topological charge +1 are

stable towards shearing since in their vicinity the amplitude of the vibrational excitations are small and vary smoothly in space, i.e., the application of a local strain will not affect significantly these vibrational modes and thus will not give rise to a plastic event. This is also the case for the boundary between two neighboring regions of $+1$ TD with opposite chirality, since also for this geometry the flow field is a smooth function in space, see Fig. 1(e). For zones having a high density of TD with topological charge -1 , the vector-field of the eigen-modes will instead be strongly modified if one applies a local strain, thus making it highly probable for a plastic event to occur. This high susceptibility can be understood by noting that the geometry of this vector-field has similarities to the motion of a particle close to a saddle-point, see Fig. 1(e). This makes that a slight perturbation in the potential energy landscape of the system will destabilize the particles in the vicinity of this TD, i.e., lead to a plastic event. We also point out that the geometry of the vector-field close to a -1 TD is very similar to the displacement field of a so-called T1 event as discussed by Falk and Langer [9], i.e., the plastic events that are considered to be relevant for the yielding of amorphous systems. This observation is thus further evidence that the TD are indeed relevant indicators for the location of PE.

When these issues are solved, I will be happy to suggest the publication of this work in *Nature Communications*.

We thank once more the referee for his/her constructive comments and hope that the modifications/responses we made convince the referee that the work is now ready for publication in *Nature Communications*.

Response to Reviewer #2

The paper “Topology of vibrational modes predicts plastic events in glasses” describes an investigation wherein the authors test the hypothesis that the plastic susceptibility of a 2D simulated purely-repulsive amorphous system can be predicted by measuring the topological defects in its vibrational eigenmodes. The idea is interesting and novel, and the paper provides valuable insights. It seems intuitive that the negative topological defects would correspond to shear-transformation zones (STZs) since these singularities have displacement fields that corresponds closely with observed STZ rearrangements (at their simplest, atomistic T1 events) and share the symmetry of the quadrupolar stress fields associated with such activity. I believe that this work is worthy of publication if some of the loose ends the authors do not adequately discuss can be addressed. Below are my questions:

We are happy to read that the referee finds our idea “interesting and novel”, and that our paper “provides valuable insights”.

1. The system that the authors choose to simulate is 2D and purely repulsive. The

sample is prepared by a quench that proceeds from $T=5$ (approx 15 times the glass transition temperature) to $T=0.1$ (about 1/3 of T_g) in about 10,000 LJ time units. To give some physical interpretation of this time-scale, for an atomistic system an LJ time unit is about a picosecond and a typical T_g might be around 600K. So, this corresponds roughly to quenching through T_g at about 300K/ns. Even by MD standards this is very high. So, these systems are in the extremely rapidly quenched (ERQ) regime. This is significant because in Ref. [15] Richard, et al. showed that predictors that are highly predictive for identifying plastic activity in ERQ glasses are not as highly predictive in deeply quenched glasses. It is particularly notable that the predictors that relied on analysis of normal modes did well in the ERQ glasses, but fared significantly less well in more stable amorphous systems, particularly in the ultra-stable regime.

EITHER The authors should be clear in their exposition about the nature of their glasses being extremely rapidly quenched. They should discuss the regime they correspond to vis-a-vis Ref. 15 and explain the limitations this implies for prediction outside the regime of ERQ glasses.

OR The authors should do further studies on a broader range of amorphous systems including ultra-stable system and include theses in the study so that the reader can be more confident that this correspondence between plastic activity and topological defects in the vibrational spectrum is not limited to the ERQ regime.

Regarding the estimation of T_g : In the Extended Data Fig. 6, we show that for this model system the mode-coupling temperature T_{MCT} is around 1.09. This value can thus be used as an approximation for the kinetic glass transition temperature T_g . So we start our quench not at 15 times T_g but at a somewhat more reasonable value of $5T_g$. In the new version of the manuscript we spell out this relevant fact.

Added text:

Line 320:

The Extended Data Fig. 5 shows that for this system the mode-coupling temperature is around $T_{MCT} = 1.09$, i.e., our initial temperature is about five times higher.

Regarding the cooling rate: We agree with the referee that the 10^7 time steps we used to quench the sample are not the maximum that can be achieved in present day simulations. However, the main goal of the present study was to propose a novel method that allows to make a connection between the plastic events at yielding and a quantity that has so far not been considered in the field of glasses, i.e., the topological defects of the vibrational modes. Of course one should not test this method for a glass sample that is not representative because it has been produced with a cooling rate that is extremely high. However, this is not really the case (and here we slightly disagree with the arguments/numbers put forward by the referee): The diffusion constants presented in Extended Data Fig. 5 shows that with $2 \cdot 10^6$ time steps one can equilibrium the current system at around 10% above the

mode-coupling temperature T_{MCT} . Numerous previous studies have demonstrated that at this temperature the liquid is in a glassy-regime, i.e., the system is no longer a normal fluid but shows all the aspects of a glass-forming liquid. The cooling time we used, 10^7 time steps, should give a kinetic glass-transition temperature that is not very far from T_{MCT} and therefore the glass state we produce is not one that is very high up in the potential energy landscape, in other words, the sample is a decent glass and certainly not an “extremely rapidly quenched glass”.

In order to address this question in a direct manner we have produced a new sample using the same protocol as the one described in the manuscript. However, this time we used 10^9 time steps to cool down the system, i.e., a cooling rate that is 100 time smaller than used before (and which is now state of the art). For this new glass sample we repeated the analysis described in the manuscript, i.e., determined the topological defects, sheared the sample (same shear rate), etc. Figure R3 shows the resulting density map for the topological defects as well as the observed plastic events. This figure is qualitatively very similar to Fig. 3e in the manuscript, i.e., one sees again a very good correlation between the -1 TD and the PE. Hence one can conclude that this correlation is not spoiled by the low cooling rate and that our results are robust.

In the new version of the manuscript we discuss this important information in more detail. In view of the point #2 of the Referee (see below), we have now added this information in a new figure of the manuscript. Please see our response to point #2 for details.

Added text:

Line 322:

The cooling rate used to produce our sample makes that the glass-former falls out of equilibrium at around T_{MCT} , i.e., at a temperature at which the dynamics is already quite sluggish. In order to make sure that the results presented for this sample are not affected in a significant manner by this cooling rate, we have repeated the analysis for a sample that has been cooled 100 times slower (10^9 time steps for the quench).

2. The authors provide evidence of good correlations between this metric and locations of plastic activity as determined via the D_{min} measure. However, the way that this correlation is reported is significantly different from Ref. 15 which compared a large number of other, possibly comparable metrics. Using some of the same means of analysis, such as reporting the correlation decay as was shown in Fig 6 of that paper would make these results comparable and increase the value of this paper greatly. It seems that it would not take much additional work.

Investigating how the correlation between PE and TD depends on the strain is definitely a good idea and we thank the referee for this suggestion. To address this point we have now measured the strain dependence of this correlation in the following manner: For a given strain γ we have identified the particles that have the top 5% of D_{min}^2 and define these particles as participants in a PE. (Thus this is the same definition we have used in

Figure R3: The spatial correlation between TD and PE in a sample generated with a cooling rate that is 100 times slower than the one for the sample presented in the main text. The strain is $\gamma = 0.025$.

the previous version of the manuscript.) We then read off from the charge field of the TD (which is independent of γ) the local charge at the location of these PE and average this charge over all the PE that are present at this γ , thus defining $\overline{C}(\gamma)$. Thus a good correlation between PE and negative TD will give a \overline{C} that is strongly negative, while a zero charge signals that there is no correlation. The γ -dependence of this average charge is shown in Fig. R4(a) (red curve). We see that at small strain the local charge density is indeed low, thus indicating a good correlation between the PE and the -1 TD. With increasing γ the charge increases but stays significantly negative up to around $\gamma = 0.075$, a point at which the sample has yielded. (The stress-strain curve is shown in panel (b).) For even large strain the value of \overline{C} is very close to 0, thus confirming that the correlation has indeed vanished once the sample has yielded.

Also included in the graph is the corresponding data for the sample generated with the smaller cooling rate (blue curve). We see that this curve is qualitatively similar to the one of the other sample, but now we recognize the presence of a plateau at intermediate values of γ (say $0.01 \leq \gamma \leq 0.05$). Thus one can conclude that for $\gamma \leq 0.01$ the proposed correlation is very high, then rises a bit but stays significantly negative and is roughly independent of strain, and finally drops to zero when the sample yields, panel (b). These results demonstrate that our indicator for the PE is indeed reliable also for other values of γ and, most noteworthy, quickly goes to zero if the system has yielded. This γ -dependence is thus qualitatively similar to the ones presented in Fig. 6 of Ref. [15], and might, for the well-annealed sample, perhaps even be more pronounced.

In order to illustrate in a more visual manner the good coincidence between the PE and the location of zones with a low TD charge, we show in Fig. R5 snapshots of the two samples at various degrees of strain.

Figure R4: **Correlation between plastic events and the topological defects as a function of strain.** (a) The average charge of the TD, \bar{C} , as a function of γ is negative for all strains less than the yielding strains. The red and blue curve corresponds to the glass samples that have been quenched with a moderate and slow cooling rate, respectively. (b) Stress-strain curve for the two glass samples. The full and dashed lines correspond to a shear in the positive and negative x -direction, respectively.

In agreement with the comment of the referee we consider this result to be very useful and therefore have included it in the new version of the manuscript. The corresponding text reads:

Line 233:

The results presented so far correspond to the strain $\gamma = 2.5\%$ and a glass sample that was generated using a moderate cooling rate. To investigate how the correlation between PE and TD depends on the applied strain we identify for a given value of γ all particles that are in the top 5% of D_{\min}^2 and calculate for these particles the value of the charge density field of the TD, thus giving an average charge $\bar{C}(\gamma)$. This average charge is strongly negative if there is a good correlation between the PE and the -1 TD, while $\bar{C} = 0$ if there is no correlation. Figure. 4 (a) shows \bar{C} as a function of γ . For the sample that has been

produced with a moderately cooling rate (red curve) the value of \overline{C} increases steadily with increasing γ , showing that the correlation between the PE and TD is high at small strain and then gradually disappears if the yielding strain is approached, see stress-strain curve in panel (b). For the sample that has been well annealed, blue curve, the value of \overline{C} at small γ is very close to the one of the first sample, showing that the quality of the correlation does not depend strongly on the nature of the glass if the strain is small. For intermediate strains the value of \overline{C} changes only weakly but once the yielding strain is approached, it rapidly goes to zero, i.e, the correlation is completely lost. Figure 5 displays the maps of the TD charge density field for the two glass samples. The different panels show the location of the PE at the indicated strains, allowing to get a visual perception of the correlation as a function of γ . Hence Figs. 4 and 5 show that the local TD charge density is an indicator that correlates indeed very well with the location of the PE, independent of the strain or the annealing of the glass.

Figure R5: **Location of the plastic events and map of the TD charge field.** Maps of the charge field of the TD, including the PE at different values of strain (stated in the panels). The filled and open symbols show the location of the plastic events when the sample is sheared in the positive and negative x -direction, respectively. Panels (a)-(c) are for the sample quenched to the glass state within 10^7 time steps and (d)-f) for the sample that was cooled more slowly (10^9 time steps).

3. The authors make the claim that “Finally we mention that the approached (sic) presented here can be generalized in a straightforward manner to three dimensional sys-

tems and it will thus be highly interesting to determine for these cases the correlation between PE and TD.” This is puzzling to the reader because topological defects can differ in significant ways in 3D from 2D. See for example (Sajedeh Afghah, Robin L. B. Selinger & Jonathan V. Selinger (2018) Visualising the crossover between 3D and 2D topological defects in nematic liquid crystals, *Liquid Crystals*, 45:13-15, 2022-2032, DOI: 10.1080/02678292.2018.1494857). There have been recent advances measuring local yield stress in 3D (<https://doi.org/10.1016/j.jmps.2021.104671>). So, describing the potential connection between these approaches would be highly valuable. However, some significant further elucidation is necessary because the generalization and connection to 3D LYS and STZ structure is far from obvious, at least to this reader.

Of course we completely agree with the referee that in general the nature of topological defects in 2D is different from the one in 3D. What we wanted to express is the simple fact that also in 3D one can calculate the eigen-vector field for each harmonic mode. Subsequently one can identify topological defects for this field (although in that case they might be lines) and hence one can again probe whether the strained sample will show plastic events that are spatially correlated with these TD. So we see no major obstacle to generalize the analysis that we have presented here to 3D. Of course in practice one might run into some technical difficulties, but overall the path is clear.

We have now modified this sentence in order to express more precisely our view on this point.

Line 301:

Finally we mention that the approach presented here can in principle be generalized in a straightforward manner to three dimensional systems since also in this case the eigen-vector field can be computed easily. Subsequently one can determine the locations of the topological defects, which in this case will be lines [48], and see whether they correlate with the position of the PE of the sheared sample. This approach might thus be an interesting alternative to other methods that have been proposed to predict the location of PE in 3D [16,49].

4. The description of how the topological defects are measured in the methods section on pages 10-11 seems unclear and possibly incorrect. The authors say, “The topological defects were then identified by integrating θ over a closed path on the lattice, and located by taking the position of the center of the smaller square with a value of 2π (-2π) of this integral, defining thus charges of +1 and -1 respectively. I believe that (a) this is not quite correct in that the authors are integrating $d\theta$, i.e. the derivative of θ w.r.t. the contour s parameterized on the interval from $[0, 1]$ with respect to ds over the contour length, and (b) they should specify that this is being done in a counter-clockwise fashion otherwise the signs are reversed. Description at this level of detail also does seem potentially relevant to the question of 3D/2D discussed above in point 3.

We thank the referee for bringing up these points. Regarding (a): We agree that one has to integrate the change of θ , i.e., $d\theta$ and not θ . This was an error in our manuscript

but the numerical calculation we did was correct. Regarding (b): We regret, but the topological charge of a defect does not depend on the way the line integral is carried out since this charge is a topological invariant. (Note that in the line integral the direction of the element ds changes if one changes the sense of the rotation, but also the sign of the $d\theta$ changes. So the line integral does not change.) We have modified this text and hope that now it is more accurate and clearer to understand. The new text reads:

Line 340:

The topological defects were then identified by calculating the line integral of $\nabla\theta$ over a closed path on the lattice giving 0 , 2π , or -2π if the loop contained no, a $+1$ TD, or a -1 TD, respectively. The location of a TD was obtained by identifying the center of the smallest square with a non-zero value of the line integral.

We once more thank the referee for his/her constructive comments and hope that the modifications/responses we made convince the referee that the work is now ready for publication in Nature Communications.

Response to Reviewer #3

The plastic flow of amorphous systems consists of series of localized rearrangements of the disordered structure (plastic events). The understanding of the relation between the plastic behavior and the amorphous structure remains elusive. In particular it is not clear to what extent it is possible to adapt the notion of defect to amorphous systems and to identify specific patterns of the disordered structure associated with plastic events.

In the context of this very active field of research, the authors of the present manuscript develop an original approach: they adapt a topological description early used to characterize (real space) structural defects to the displacement field associated to low frequency vibrational modes.

In particular, the authors show convincingly the correlation between the location of negatively charged topological defects of the vibrational modes of a 2D model glass and the location of plastic events obtained by shearing (backward and forward) the same amorphous system.

The manuscript is very well written. The results are novel, interesting and stimulating. I think that the manuscript can be published in Nature Communications provided the authors consider the following points.

We thank Dr. Vandembroucq for the very positive evaluation of our work and are pleased to read that he liked the manuscript and the presented results, and acknowledges our approach to be original.

The approach that the authors develop for the vibrational modes is reminiscent of early attempts of identifying topological defects in the real space amorphous structure (discli-

nation, frustration, etc.). This is very briefly mentioned in the abstract/introduction but the authors may want to emphasize and discuss this point in a more explicit way in their manuscript.

This is indeed a useful suggestion, also made by Referee #1 (point 1). We thus copy here the new text on this:

Line 74:

...topological properties. The idea that such properties might be useful to understand certain thermodynamic and kinetic features of glasses has been suggested only in recent years [44,45]. Subsequently Baggioli et al. have put forward the relevance of topological properties by showing that plasticity is mediated by topological features in the non-affine displacement field of glasses under deformation [39]. In the present work we find that...

A limit of the present study is that the tensorial character of the plastic behavior is not fully taken into account. Depending on the direction chosen to shear the system, different plastic events can actually be unveiled. When shearing along only one direction (even backward and forward) one gets only a subset of the potential plastic events. I think the authors should at least use the other shear direction. This would allow them to reveal if not all, at least a significant additional population of plastic events and this will likely improve the level of correlation they get between topological defects and plastic events.

We thank the referee for this good suggestion. Following his advice, we have now sheared our sample in the orthogonal y -direction to take the tensorial character of the PE's into account. The resulting location of the PE's are shown in Fig. R6b and superposed to the map of the charge density of the TD's (which is independent of the shear direction since it is obtained in the non-sheared sample). For the sake of comparison we reproduce in Fig. R6a the corresponding graph when the system is sheared in the x -direction (this was panel (e) of Fig. 3 in the original version of the manuscript). One sees that the two shear directions do indeed produce PE's that are spatially very close together, which indicates that the direction of shear is not that important, at least not for the system considered here. Since we consider this to be an interesting insight, we have added Fig. R4 as Extended Data Fig. 8 of the new version of the manuscript and mention these finding in the main text.

Line 218:

(We also mention that shearing the system in the y -direction gives rise to PE's that are spatially strongly correlated with the ones shown in panel (e), indicating that the direction of the shear does not play a very important role in our case, see Extended Data Fig. 8.)

We once more thank Dr. Vandembroucq for his useful comments and hope that the modifications/responses we made convince him that the work is now ready for publication in Nature Communications.

Figure R6: **Charge density field of the TDs (color map) and of the PE's (symbols) at strain $\gamma = 0.025$.** Panel (a) shows the PE's resulting from a simple shear parallel to the x -direction and in panel (b) the ones for shear parallel to the y -direction. Filled and open symbols correspond to the positive and negative direction, respectively.

REVIEWERS' COMMENTS

Reviewer #1 (Remarks to the Author):

The Authors have carefully considered my questions and addressed all my previous issues. They have also added additional analysis and more in detail explanations regarding the results.

As I have already argued in my previous report, I believe that this is a nice piece of work, and it should be now published in Nature Communications in its current form.

Reviewer #2 (Remarks to the Author):

Reiterating my prior review, I find this manuscript to be of high value and I support its publication now that the authors have addressed all of the issues I raised.

Reviewer #3 (Remarks to the Author):

I am fully satisfied with the detailed response of the authors and the changes they made in the manuscript.

I note below a few typos/unclear writings to be checked by the authors. I also add a short comment.

l54: "it is reasonable the attempt THE correlate" -> "it is reasonable the attempt TO correlate" ?

l118: "this RATIONALIZING" -> "this RATIONALIZES" ?

Flg. 3, 6th line of the caption: "(see legend)" -> "(see text)" ?

A short comment. I find it very interesting that, as mentioned by the author in the discussion, the correlation between PE and TD can exist only if the sample can be locally considered as an elastic medium. This is very consistent with what is obtained for the local yield stress method. There the best results are obtained with patches whose size is such that continuum mechanics is reached (diameter about 10 interatomic distances) [Patinet et al PRL 16].